# Is Australia Neglecting the Local Topography When It Comes to Catastrophic Costs and Ending Tuberculosis?

**DOI:** 10.3390/tropicalmed3040126

**Published:** 2018-12-19

**Authors:** Krista Watts

**Affiliations:** Victorian Tuberculosis Program, Melbourne Health, Parkville 3050, Australia; krista.watts@mh.org.au; Tel.: +61-393-429-476

**Keywords:** tuberculosis, socio-economic, catastrophic costs, financial

## Abstract

Efforts to eliminate tuberculosis as a public health problem require reductions in mortality, incidence, and the eradication of associated catastrophic costs; however, the question of catastrophic costs is often neglected, particularly in the context of low-incidence settings like Australia. This study reviews the financial support provided to those identified as in need, and in receipt, of economic aid from the Victorian Tuberculosis Program. The study design used Epstein’s clinical data mining framework to produce descriptive statistics which were supplemented by clinical collaboration. A consistent one-third of those receiving care from the Program due to a notification of active tuberculosis received emergency financial relief over the study period. Overwhelmingly, funds were used to relieve financial distress, and each year approximately one-third of the expenditure was used to support 2% of those people notified as affected by tuberculosis (or 7–9% of those in receipt of funds). Many of this 2% experienced income loss and expenditure that may be considered catastrophic. Further investigation is needed to better define and understand the nature of catastrophic costs in the context of universal health care and existing low tuberculosis incidence.

## 1. Introduction

History shows that declines in tuberculosis rates are possible through improved socio-economic development; with the fastest decline in recorded history being in the context of universal health care, rapid socio-economic development and the availability of effective medications in the middle of the 1900s [1,2]. These circumstances have contributed to long-term stable tuberculosis incidence rates in many highly developed settings, including Australia. A shared vision of a world free of tuberculosis builds upon these components with milestones and targets adapted globally to reduce the number of tuberculosis deaths, the incidence rate of tuberculosis, and to have zero tuberculosis affected families experience catastrophic costs due to tuberculosis [3]. The elimination of catastrophic costs is a mainstay of the End TB Strategy, yet its study, especially in low incidence tuberculosis settings, is thin. The local topography in most low incidence settings includes universal health care and socio-economic protections; this should not invite neglect of this SDG. Understanding the topography enables the judicious application of the roadmap provided by the End TB Strategy.

Australia has worked to secure a tuberculosis incidence rate of 6.8 per 10,000 population and 87% treatment coverage for 2017 [4]. Key to this has been the development of a national framework for tuberculosis management working to “minimize the national burden and human impact” of tuberculosis [5]. Tuberculosis care is provided in the context of universal health care - free medication and appointments for those diagnosed with active tuberculosis.

Socio-economic development has been strong in Australia; however recent data indicates a slowing of wage growth and household income for the cost of living in the current decade [6]. Social security, in the form of welfare allowances and pensions, are provided by the Commonwealth Government and administered through Centrelink to those eligible. Financial, food, and housing insecurity are present [6,7,8] in Victoria. Program experience indicates that financial hardship is a barrier to engagement with tuberculosis testing and treatment in the Victorian setting. The anecdotal evidence speaks of hidden and indirect costs associated with tuberculosis such as loss of income and non-medical costs pre-diagnosis, at diagnosis, and during the intensive phases of treatment in particular. These costs are not unique to the setting [9,10,11,12,13,14].

The Victorian Tuberculosis Program is a centralized agency of tuberculosis care, provided via casework and inclusive of contact tracing and case finding. Partnering with specialist clinical and laboratory services, the Program works towards patient-centred care to optimize engagement with treatment to cure by emphasizing tailored self-administration of tuberculosis medications. The Program assists those facing food, income, and housing insecurities to access mainstream services and have been able to provide limited emergency financial relief in the form of vouchers.

There is little systematic data on financial insecurity and the use of socio-economic relief in low tuberculosis incidence settings. This project reviews the Program’s financial support for those affected by tuberculosis in order to gain insight into the nature and extent of financial insecurity. In the absence of a costing survey, it asks the question of whether Australia’s second largest state and “an industrialised setting with universal health care and low TB incidence” [15] (p. 550), and both government and non-government social protections, is alert to local topography and ready to engage the 2020 SDG milestone of *no* households experiencing catastrophic costs due to tuberculosis?

## 2. Study Design

To conduct a descriptive monitoring study, available agency data was accessed for practice-based research consistent with the principles of clinical data-mining (CDM) as per Epstein [16]. Catastrophic costs are defined as direct medical and non-medical costs and income losses that sum to 20% or more of household income [9] (p. 138) that is associated with tuberculosis.

This project uses existing data that had been previously provided for the acquittal of funds and as an opportunity for the Program to engage in an annual audit of service provision with opportunity for practice discussion to illuminate socio-demographic, economic, and clinical factors that may be associated with higher percentage patient costs. It begins to provide baseline data of the experiences of Victorian households financial well-being due to tuberculosis and encourages early recognition and preventative action regarding the loss of social capital.

Data on the receipt and dissemination of socio-economic supports in the form of grocery vouchers has been held by the Program in a systematic manner since 2007. Vouchers have been funded to aid and help those suffering from tuberculosis. Vouchers are purchased in bulk in $20 and $50 denominations. Acquittal is made per the calendar year.

In 2014, financial acquittal requirements and practice questions invited a review of recording methods at which time a Microsoft EXCEL^®^ spreadsheet was employed to document core information about the dissemination of grocery vouchers. Each time a patient is assessed by a staff member as likely to benefit from the provision of a voucher they must document the date, recipient’s unique record number, the voucher’s unique identifier, the amount received in total and (using pre-determined categories) explain the clinical reason necessitating the voucher. Pre-determined clinical reason categories were defined by the work of van den Hoff et al. [17] and practitioner consensus (see Appendix A for definitions of clinical reason). By recording incoming and outgoing voucher identifiers the Program is able to provide responsible reporting to funding bodies.

### Methods

The voucher dissemination data is matched with demographic and epidemiological data from the Public Health Event Surveillance System (PHESS) based on recipient unique record number. This is a subset of the PHESS surveillance data for all people diagnosed with active tuberculosis in Victoria. In this way the data-set captures all people residing in Victoria diagnosed with active tuberculosis, and who have received a voucher from the Program. This data-set is built and reviewed annually with regard to reporting against funding agreements. This provides for a rich, de-identified, data-set informing practice knowledge development, clinical decision-making, and practitioner reflection.

Data is manually cleaned and then analyzed using Microsoft EXCEL^®^, giving consideration to descriptive statistical analysis with a focus on annual spend, central tendency, range, data accuracy and completion, disbursement patterns including by sex, practitioner, time and clinical reason. Results are reported back to Program staff annually and discussed with regard to implications for practice and program development; providing clinical collaboration. Particular attention is given to the psychosocial circumstances of those individuals who receive the highest ten total amounts for the period. This collaborative discussion operates as a validity check on the data and encourages the team to reflect on operations, practice, and context.

As data in this study were collected and used under the legislative authority of the Public Health and Wellbeing Act 2008, or for acquittal against funding agreements, approval from a Human Research Ethics Committee was not required under the rules of our institutions.

## 3. Results

This project reports on three years of data, 2015–2017 inclusive. Recognizing that reporting efforts and nominal data definitions have evolved since the 2015 data examination, the annual analysis demonstrates surprising similarities and consistency across key data points. During the study period, a consistent one-third (see Table 1) of those individuals notified with active TB received emergency funds in the form of vouchers from the Program. This was the case regardless of increases in notifications across the same period.

The distribution of sex across the recipients, by year, was not markedly different by sex or from surveillance data (52% male in 2015, 52% male in 2016, 51% male in 2017) suggesting that there is no sex-based bias in disbursement or need for emergency financial relief from the Program.

As noted in Table 1 the most common interval from notification of tuberculosis disease to receipt of a first voucher is 140 days for 2015, 8 for 2016, and 288 for 2017. A highlight of Figure 1 is a contact testing outlier that is the likely cause for the 2017 number. A bimodal analysis shows a mode of 42 days in the second position for 2017. The typical interval between the Program receiving a notification of a new diagnosis of tuberculosis disease and that person receiving a voucher is illustrated in Figure 1. The green lines in Figure 1 mark milestones in treatment: the end of the intensive phase (week 8), the conclusion of a six month course of treatment (week 26), and the conclusion of a nine-month course of treatment (week 39), and 12 month course of treatment (week 52). Each calendar year demonstrates similar patterns with the provision of vouchers being most frequent in the earlier stages of treatment and tapering off as treatment advances.

Review of the 2015 data, and a request to track funds against each voucher as a singular item, altered recording for the following years, making it possible to consider the absolute value of vouchers and the average value of vouchers, by month, against the notification rate, by month. On the whole, the mean and mode data on the period from notification to voucher receipt shows voucher dissemination followed notification trends (notice the green line and the black trend line in Figure 2 and Figure 3). Exceptions would appear to be present with clinical collaboration reporting that fluctuations in the value of vouchers provided reflects voucher availability, seasonal expenses, and case complexity. The 2016 data indicates a higher average value of vouchers per notification for the first quarter. It is likely that this data is compounded by individuals notified in 2015 still being active for this period.

The three years of the project has evidenced significant shifts in financial need across the number of notifications. The lowest amount received by any one recipient was $20 with the highest values ranging from $880 in 2015, $1410 in 2016, to $1960 in 2017 (Table 1). This is consistent with data relating to occasions of service with a range of one occasion (at $20), through to 36 occasions in 2017, 32 in 2016, and 16 in 2015 (Table 1). The 7–8% of cases per year in receipt of more than $500 represents approximately 2% of notified cases per year across the study period. People most often receive between $30 and $50 or $100 and $190. The most common clinical reason for dispensing funds is recorded as financial relief.

In 2015 most (92%) recipients received between $20 and $390 over the 12 month period with dual modes in the $30–50 and $100–190 groups. Most funds (40%) were spent to relieve financial distress, as the first year of data collection 37% of funds were not allocated a nominal clinical reason for disbursement, 13% was provided as an incentive, 9% to reward, and 6% was provided to redress issues of social justice.

The recipient of vouchers at the highest monetary range received 5% of the total spend of vouchers for the year, and the top 10 recipients combined received 33% of the total voucher spend (range 2–5% per recipient). Of this spend, 16% was identified as being spent on financial relief and 5% for reward and incentive combined. Being the first year of reporting in this manner, the remainder of the expenditure for these recipients is not nominally labelled.

In 2016, an individual recipient received over $1000 with most recipients (92%) in receipt of less than $500. Dual modes are present in the $30–50 and $100–190 groups. Of the 2016 spend, 64% was used for financial relief, 17% as an incentive, 15% for social justice redress, and 4% for reward.

The total percentage of voucher dollars spent on the 10 recipients who received the highest amounts was 30% of the total. The individual recipient who received more than $1000 received 7% of the total voucher expenditure for 2016 all of which was named as an incentive. The second highest recipient received 4% (range 2–7%). Of the top 10 recipients 16% of funds were allocated due to financial relief, 9% as an incentive, 6% for social justice reasons, and 1% as a reward.

In 2017, most (93%) recipients received less than $500 with dual modes in the $30–50 and $100–190 groups; 80% of the total spend was used for financial relief, 9% for incentive, 6% to redress social justice issues, and 3% to reward individuals for their engagement in treatment.

Across the 10 recipients receiving the highest amounts 32% was allocated on the grounds of financial relief, 5% for the purpose of incentive, 2% for a non-categorical purpose (primarily the purchase of portable heating and cooling relating to housing and health issues), and <1% to redress social justice issues. Again, one recipient was given more than $1000. This recipient received 9% of the total spend for the period, all of which was noted to be for the purpose of financial relief. The recipient in possession of the second highest value received 4% (range 2–9%), with the top 10 values making up 37% of the total spend on vouchers for the year.

Exploring the demographic and epidemiological data of the recipients to receive the 10 highest values across the study period demonstrates increasing psychosocial complexity. Consistent with notification data seven to nine of the 10 each year are overseas born. Of the six Australian born recipients identified across the study period, half report one or more parent as being born overseas. This is consistent with the 2016 Victorian census data which notes that unlike the typical Australian, the typical Victorian reports at least one overseas-born parent [18]. The majority of Australia’s tuberculosis notifications are for individuals who identify as over-seas born [19].

Since 2015 the representation by sex has shifted from eight of the 10 to half of the 10 being male. Homelessness, or insecure housing, has been a consistent concern for half of the group. Insecure income has become an increasing concern, with four of the 10 in 2015 to seven of 10 in 2017 describing cash-based employment, casual employment or underemployment. For the purposes of the study, Social Security allowances and pensions have been included in a single category and considered secure. Allowances are only secure if an individual meets their Social Security obligations. Psychosocial issues can compromise this security; two-thirds of this group disclosed a compounding complexity such as a dual diagnosis, alcohol and other drug (AOD) use, mental health, forensic history and/or family breakdown which can make it difficult for individuals to meet the requirements of conditional payments. Visa status and its impact on welfare eligibility has become a greater concern in 2016 and 2017. Individuals affected by MDR-TB are also well represented within these top ten recipients, reflecting an increase in MDR-TB diagnosis in Victoria in recent years. See Appendix B for a summary.

## 4. Discussion

As aforementioned, this project has been a means of reviewing Program practices and providing acquittal of fund disbursement on an annual basis. In comparing annual data obvious patterns begin to appear detailing local topography with regard to the financial costs of tuberculosis. Across the study period, one-third of those receiving care from the Program due to a notification of active tuberculosis receive emergency financial relief from the Program. Regardless of increasing numbers of notifications, the figure has been consistent. This has funding and sustainability implications.

Overwhelmingly funds are used to relieve financial distress; 40–80% of expenditure is reportedly due to observable financial distress. Most people receive funds earlier in treatment which is reflected in the consistent dual modes of $30 to $50 and $100 to $190 seen annually. Clinical collaboration discussions show that this is likely to reflect loss of income and non-direct medical costs or costs of delayed diagnosis. Further investigation into patient costs and Program practice to develop a comprehensive understanding is necessary.

Each year approximately 30–37% of the expenditure is used to support 2% of those people notified as affected by tuberculosis (or 7–9% of those in receipt of funds). Psychosocial and medical complexity would appear to be significant contributors as annually there is a 2% increase in the value of funds provided to the recipient of the most funds.

These figures suggest that in a developed setting with universal health care and low tuberculosis incidence, approximately 30% of households affected by tuberculosis are experiencing financial distress. Clinical collaborative discussion submits that recipients receiving the highest amounts (2% to 9%) may be experiencing catastrophic costs due to tuberculosis. This project operates as a proxy only against the recommended definition of catastrophic costs as income loss, or costs, of greater than 20% of annual income. In considering the nature of CDM it has been useful to employ supplementary processes of collaborative data analysis whereby staff directly responsible for assessment and provision of vouchers have had an opportunity to discuss the results on an annual basis. These discussions highlight the direct and indirect costs of tuberculosis diagnosis and treatment holistically and in context.

As with other settings [11], the perceived relevance and value of tuberculosis testing within the context of a low incidence setting and accessibility to diagnostic testing are likely to impact upon pre-diagnostic costs. Clinical collaboration asks that we consider the child with meningeal tuberculosis, whose parents require extended periods of personal leave during a lengthy delay in diagnosis, resulting in the loss of casual employment and the exhaustion of all leave entitlement prior to diagnosis. This delay is directly related to an understanding that the child is Australian born and that tuberculosis is not a significant concern, despite the clinical presentation or the parent’s country of origin. At diagnosis, the parents make an application to Centrelink for Social Security payments and face lengthy application and processing times, resulting in further income loss and increasing debt; employment is lost. Once in receipt of Social Security the household is living on approximately one-third of their previous income; a catastrophic loss.

The working single parent who experiences lengthy periods of misdiagnosis and ill health prior to diagnosis is also considered. The single mother is understood to be in receipt of payments that leave her under the poverty line [8] and unable to keep up with the costs of living in Victoria. Her health compromises her capacity to meet the conditions of Centrelink payments, resulting in the ceasing of income. Without income, she accrues debts and is reliant upon emergency relief for food and advocacy with her landlord to retain accommodation.

Post-diagnostic costs reflect the organization of tuberculosis care and accessibility to socio-economic supports. For example, in rural and remote Victoria patients may travel hours to access specialist tuberculosis care, accruing transport costs and loss of income. Households located in these settings may be more vulnerable due to drought or other environmental factors such as funding contracts with private health care providers by State-run organizations, or due to increased psychosocial vulnerabilities as highlighted in collaborative discussion. Australia’s largest provider of hunger relief (Foodbank) highlights that four million Australians experienced food insecurity in 2017 and that those living rurally and remotely are more likely to experience food insecurity [7]. The Foodbank report that emergency relief providers are unable to assist approximately 7% of those who approach them, that 30% of those experiencing food insecurity attribute it to living on a low income or Social Security benefits, and that more than half of those living with food insecurity spend more than 20% of their household income on food (the average Australian household spends 10% of their income on food) [7]. Current screening practices invite Program staff to consider a household’s primary source of income in recognition of the conditional nature of Social Security payments and consideration of its worth against the cost of living, however early screening of a household’s expenditure using a formula such as household expenditure by percentage for food and housing costs may prove more helpful. Assessment of this requires further research in the setting and capacity building to facilitate implementation and evaluation.

In examining the data on interval from the Program’s receipt of notification of an individual’s diagnosis with active tuberculosis to their first receipt of a voucher, it is apparent that financial need typically drops off during the period of treatment, presumably as people return to work or other sources of income such as Social Security payments commence. For others, this data suggests that tuberculosis-specific emergency financial relief is needed for the duration of treatment. From collaborative discussions and the analysis of psychosocial data related to those top fund recipients, those benefiting from long-term specific financial relief would appear to be those who are most vulnerable to lost income, accrued debts, and face non-medical costs related to tuberculosis. Further study is needed to strengthen understanding and facilitate tailored responses for these households in order to halt the loss of social capital.

Interrogating the data further, a sharp rise across all three years at week two to three of treatment, when a person is typically discharged home from a period of hospital isolation, indicates that many of those in receipt of vouchers benefit most from them in returning to activities of daily living.

Clinical collaborative discussions suggest that the figures evidencing an interval of greater than 12 months between notification and receipt of a voucher may reflect MDR-TB, diagnosis in a previous calendar year (which impacts upon the calculation), relapse, or complexity and need in transitioning into alternate socio-economic supports and ineligibility for these supports.

Illustrated in Program collaborative discussions and data relating to social determinants of health with regard to the circumstances of the individuals receiving the highest values in vouchers (typically over extended periods) is the need for tuberculosis-specific emergency relief. Collaborative discussions speak to the context of a tailored welfare system that requires lengthy application via an increasingly English language web-based system. For those eligible and able it highlights the importance of early vouchers in assisting households to keep food on their tables whilst applications are processed. In those situations in which individuals are not eligible for welfare services, or where welfare services are too slow and the drop in income too significant to prevent catastrophic costs occurring, vouchers are critical. These stories tell of families using vouchers to purchase a single serving of rice and protein to feed a household of four as their Social Security allowance was used for accommodation costs. Rough calculations based on the epidemiological data and practice knowledge suggests that the income of these families is often in the lowest 40% of the population and their accommodation costs are often 30% or more of their new income, leaving them vulnerable to housing insecurity. Individuals ineligible for Social Security are typically those who are undocumented, on dependent visa types, or a range of visa types with limited rights e.g., bridging visa, New Zealand citizens (see Appendix B). The CDM approach taken in this study interrogates the social determinants of health for the top ten recipients of emergency relief funds only leaving an invitation to examine the lived circumstances of *all* persons affected by tuberculosis.

The Program updated its business rules at the review of 2016 data determining that all vouchers are allocated to the notified individual regardless of whom they were given to e.g., if a child is notified as having tuberculosis all vouchers are recorded against the child’s unique identifier although they are given to a guardian, vouchers given to a contact are recorded against the notification record. This decision had a significant influence on the 2017 measures of central tendency for days from notification to first voucher receipt as a number of contacts were provided with vouchers as incentives to test for tuberculosis (on the same day) in the context of homelessness. This value is noted on Figure 1 as contact tracing.

Spikes in the value of the vouchers given appear consistently at the close of winter in Victoria and in the average value of vouchers provided as school terms commence in February/March. Staff movement and the psychosocial complexity of the lives of individuals are also noted in the collaborative discussion as likely contributors to rises such as those observable in June 2016 and August 2017.

Current survey models supported by the World Health Organization demonstrate a connection between service delivery models and the distribution of costs across the categories of direct non-medical, income loss, and direct medical [9]. The clinical collaborative discussion suggests that the Victorian experience is one of significant income loss followed by direct non-medical costs. The data to support this and to inform models of care is thin. Further research will enable consideration of patient-centred care models that reflect the lived experiences of those affected by tuberculosis and support the re-establishment of local action towards partnerships and funding to improve socio-economic protection for households.

Catastrophic cost for those with tuberculosis and their households is a central measure in the End TB Strategy, and an indicator under the SDG, that at this time is not adequately measured in low-incidence settings. This project illustrates a need for timely access to social protections (tuberculosis specific or otherwise) and the measurement of household costs (direct, indirect, and lost) for those affected by tuberculosis not only as a matter of practice and financial accountability, but as a human right to ensure that accessing tuberculosis diagnosis and treatment does not cause financial hardship. In short, the Victorian topography requires further study in order to best apply the End TB Strategy roadmap.

As with most CDM projects, this one is only as strong as the data available. Reporting on missing data shows that although rates are low, they are not absent and this impacts upon the project as a whole. Furthermore, surveillance data and funding data is not designed to illuminate the nuances of social determinants of health nor do they define catastrophic costs.

What this work does highlight is that regardless of the wealth of the setting patients and their families, experiencing socio-economic disadvantage as a result of illness with, and treatment for, tuberculosis and that socio-economic supports are perceived by Program employees as a core component of the work as it optimizes treatment outcomes. The project invites further tailored research into measures of socio-economic disadvantage and the objective of no tuberculosis affected household experiencing catastrophic costs related to tuberculosis regardless of the economic setting or tuberculosis incidence rate. This research must give consideration to the tuberculosis specific SDG indicators which incorporate direct non-medical, indirect costs, and loss of income, in addition to direct medical payments, for diagnosis and treatment.

## 5. Conclusions

Australia must stop neglecting the local topography if it is to effectively implement the End TB Strategy roadmap, especially with regard to catastrophic costs. Milestones and targets cannot be measured without baseline data. Indicators to progress cannot be measured in the absence of contextual definition.

Tuberculosis treatment in Australia is free; however, proxy markers indicate that the supplementary and non-medical payments and the associated indirect costs accrued by those with tuberculosis can have a significant, arguably catastrophic, impact upon tuberculosis affected households in Victoria. As global elimination of tuberculosis will not occur without all countries achieving the End TB Strategy vision, and as a matter of human rights, further targeted survey is required in the setting to halt the costs associated with tuberculosis. Survey should include patient cost, including those related to diagnosis, treatment, loss of income, and accrued debt, and treatment outcome in the context of universal health care. This approach enables the development of baseline and periodic measure against the milestones and for the documentation of the magnitude and drivers of costs incurred by patients and their households. This information can then be used to guide responses and optimize engagement with treatment locally for international contribution.

## Figures and Tables

**Figure 1 tropicalmed-03-00126-f001:**
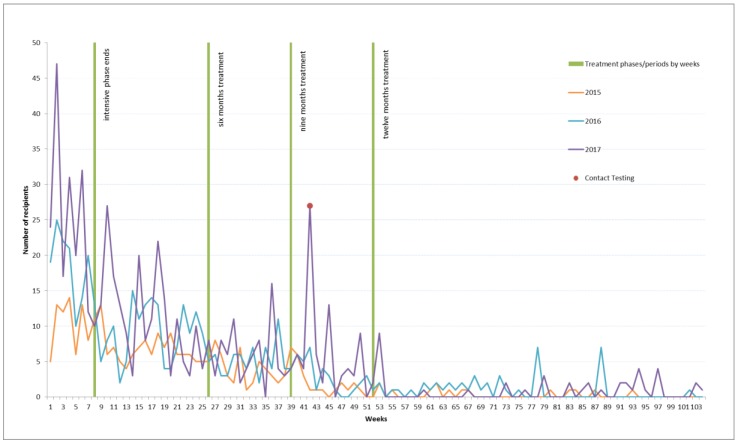
Days between Notification and Receipt of First Voucher by Year, 2015–2017.

**Figure 2 tropicalmed-03-00126-f002:**
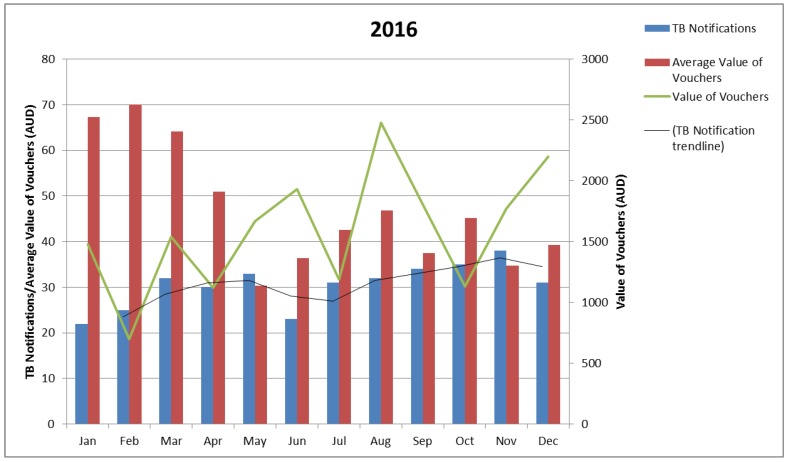
Value of Vouchers Provided by Month, 2016.

**Figure 3 tropicalmed-03-00126-f003:**
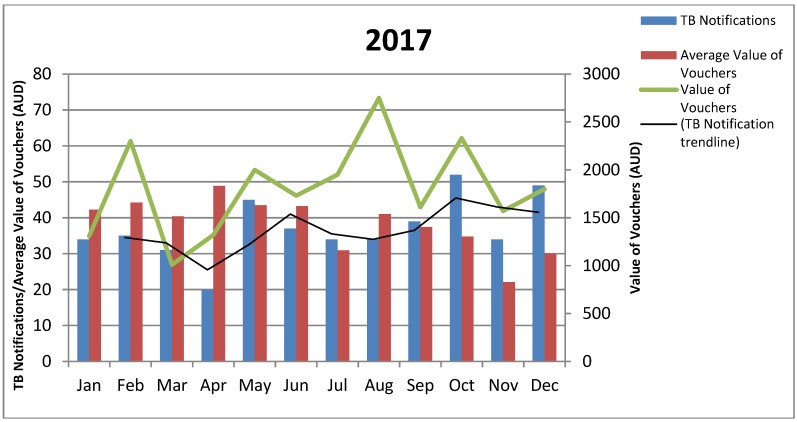
Value of Vouchers Provided by Month, 2017.

**Table 1 tropicalmed-03-00126-t001:** Voucher Dissemination Summary by Year, 2015–2017.

Descriptive Data Type	2015	2016	2017
Cases in receipt of vouchers (% of total notifications)	112 (32%)	119 (32%)	131 (29%)
Total number of vouchers provided	N/A *	475	592
Occasions of service–total count by date	288	296	355
Most common value given per case (range)	$50 ($20–$880)	$50 ($20–$1410)	$50 ($20–$1960)
Median voucher value per case (IQR)	$100 ($50–200)	$100 ($50–200)	$100 ($50–190)
Most common number of occasions of service per case (range)	1 (1–16)	1 (1–32)	1 (1–36)
Median occasions of service (IQR)	1 (1–3)	2 (1–3)	2 (1–5)
Sex of voucher recipient (% of total recipients)	40%♀ 60%♂	48%♀ 52%♂	43%♀ 57%♂
Common interval by days from event date to first voucher	-	-	-
Median (range)	120 (1–647)	119 (2–709)	106 (0–773)
Mode (IQR)	140 (50–196)	8 (43–231)	288 (39–246)

* vouchers were not uniquely identified in early data 2015.

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
