# Peer review of "Is Australia Neglecting the Local Topography When It Comes to Catastrophic Costs and Ending Tuberculosis?"

_tropicalmed, 2018, doi:10.3390/tropicalmed3040126_

Round 1
Reviewer 1 Report
The author uses the word 'however' a lot in some sections of the article. Reducing the frequency of the use of this word would greatly enhance the flow of the article.
The legends of figures 2 and 3 could be improved for greater clarity (for example, there are two Y axes and it is unclear which one corresponds to the line graph and which one to the histogram bars). The distinction between "Value of Vouchers" and "average value of vouchers" thus becomes confusing.
Line 274 page 9, "Discussions speak to..." needs a citation/reference.
Phrases such as "As noted elsewhere" perhaps need to be justified by citations (unless the author meant, "as aforementioned").
The discussion raises a number of interesting threads that deserve to be represented in the literature on TB care and prevention. Would the author be able to signpost their key points or perhaps use subheadings to outline the themes being discussed?
The author invites future research. I believe that the author can go one step further and outline the kinds of future research that would be fruitful. The shortcomings of this research project are completely understandable given the ethical and methodological constraints on the project-- highlighting these constraints will allow the author to provide more exegesis about dimensions of TB care and prevention not captured in the current study. Indeed, the research points to fascinating future qualitative studies that could investigate numerous angles such as the decision-making of healthcare professionals behind offering vouchers for financial relief, incentive, social justice redress, or reward--which would provide much needed insight into the experience of TB patients in a low-burden, high-income setting.
The conclusion could be expanded if the author wished.
Reviewer 2 Report
Abstract: - include the name of the country where the study was developed Introduction: - The introduction finishes with this phrase: "It asks the question of whether Australia’s second largest state and “an industrialised setting with universal health care and low TB incidence”, and both government and non-government social protections, is on track to make 2020 SDG milestone of no households experiencing catastrophic costs due to tuberculosis?" The study does not investigate catastrophic costs or whether the financial support received by TB patients is enough to cover patient's costs. Thus, it is not possible to answer the question stated in the introduction. Methods: - The section brings the concept of catastrophic cost, but it is not clear how this data was collected. Was there any costing survey in the country to assess the incidence of catastrophic costs among TB patients? Results: - Report the number of TB cases notified each year in Table 1 - Identify the meaning of green lines in figure 1 - Figures 2 and 3 are confusing, lot's of information and no identification of axis "y". Also, I could not understand the meaning of the black line - Figure 1 says "average vlaves of vouchers. I think the correct is "average value of vouchers" - It would be interesting to show a table with socio-economic characteristics of the voucher recipients and type of TB (drug sensitive, multidrug resistant, new cases, relapse cases, retreatment, HIV co-infection) Discussion - The section brings some hypothetical scenarios, make assumptions about catastrophic cost, financial distress. However, the results of the study do not support these ideas. Thus, it seems out of context. - The author suggests that voucher recipients can face catastrophic costs and financial distress. However, the study does not investigate patient's costs or incidence of catastrophic costs among the recipients. Thus, it is not possible to make these assumptions - The author also make other assumptions about direct and indirect costs (paragraph 218), pre-diagnosis costs (paragraph 223), post-diagnostic costs (239). however, the study does not bring any costing data and there is no mention to previous TB costing survey in the country. - Are paragraph 223 and 223 based on a hypothetical scenarios? I there any evidence to support these assumptions?
Author Response
Please find attached responses to reviewer two

Round 2
Reviewer 2 Report
Figure 2, label for red bar is "Average Valve of Vouchers"?
Author Response
It should be "Average Value of Vouchers". Thanks.